

# Characterizing transcriptomic responses to sediment stress across location and morphology in reef-building corals

Jill Ashey[1], Hailey McKelvie[2], John Freeman[2], Polina Shpilker[2], Lauren H. Zane[1], Danielle M. Becker[1], Lenore Cowen[2], Robert H. Richmond[3], Valerie J. Paul[4], Francois O. Seneca[5] and Hollie M. Putnam[1]

[1] Department of Biological Sciences, University of Rhode Island, Kingston, Rhode Island, United States
[2] Department of Computer Science, Tufts University, Medford, Massachusetts, United States
[3] Kewalo Marine Lab, University of Hawaii at Manoa, Honolulu, Hawaii, United States
[4] Smithsonian Marine Station, Smithsonian, Fort Pierce, Florida, United States
[5] Centre Scientifique de Monaco, Monaco, Monaco

Corresponding author
Jill Ashey, jillashey@uri.edu

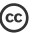

## ABSTRACT

Anthropogenic activities increase sediment suspended in the water column and deposition on reefs can be largely dependent on colony morphology. Massive and plating corals have a high capacity to trap sediments, and active removal mechanisms can be energetically costly. Branching corals trap less sediment but are more susceptible to light limitation caused by suspended sediment. Despite deleterious effects of sediments on corals, few studies have examined the molecular response of corals with different morphological characteristics to sediment stress. To address this knowledge gap, this study assessed the transcriptomic responses of branching and massive corals in Florida and Hawai'i to varying levels of sediment exposure. Gene expression analysis revealed a molecular responsiveness to sediments across species and sites. Differential Gene Expression followed by Gene Ontology (GO) enrichment analysis identified that branching corals had the largest transcriptomic response to sediments, in developmental processes and metabolism, while significantly enriched GO terms were highly variable between massive corals, despite similar morphologies. Comparison of DEGs within orthogroups revealed that while all corals had DEGs in response to sediment, there was not a concerted gene set response by morphology or location. These findings illuminate the species specificity and genetic basis underlying coral susceptibility to sediments.

# INTRODUCTION

Coral reefs are incredibly diverse marine ecosystems, providing numerous ecological and economic services such as biodiversity, cultural value, coastal protection, fisheries, and tourism (*Reaka-Kudla, 1997*; *Sumaila & Cisneros-Montemayor, 2010*; *Costanza et al., 2014*). Reef-building corals form a critical nutritional symbiotic relationship with unicellular photosynthetic algal endosymbionts in the family Symbiodiniaceae

(*LaJeunesse et al., 2018*). The carbohydrates produced by the algal photosynthesis are translocated to the coral to be used as its primary energy source, supporting the daily respiratory carbon demand of tropical corals (*Muscatine & Porter, 1977*; *Muscatine et al., 1984*). This coral-algal symbiosis fuels reef productivity and accretion (*Roth, 2014*), but is sensitive to changing environmental conditions that can impact the symbiosis such as light, nutrients, temperature, pH, and sediment (*Hoegh-Guldberg et al., 2007*; *Davy, Allemand & Weis, 2012*). For example, under exposure to sedimentation, or the downward fall of sediment from the water column toward the benthos (*Schlaefer, Tebbett & Bellwood, 2021*), corals display reduced photosynthetic efficiency (*Weber, Lott & Fabricius, 2006*; *Rushmore, Ross & Fogarty, 2021*), increased respiration rates (*Riegl & Branch, 1995*; *Browne et al., 2014*), decreased calcification (*Bak, 1978*), and rapid consumption of energy reserves (*Sheridan et al., 2014*). While sediment transport naturally occurs on reefs, suspended sediment caused by anthropogenic activities such as dredging, runoff, and coastal development have increased (*Rogers, 1990*; *Fabricius, 2005*; *Erftemeijer et al., 2012*; *Miller et al., 2016*; *Cunning et al., 2019*).

Deposited sediment and suspended sediment are the two primary ways that sediment interacts with corals (*Rogers, 1990*; *Fabricius, 2005*; *Erftemeijer et al., 2012*). Deposited sediment occurs when sediment particles settle directly on the coral surface, making physical contact with the tissue. Passive removal of sediment includes gravity or flow removing it from the coral surface (*Lasker, 1980*; *Jones, Fisher & Bessell-Browne, 2019*). In response to deposited sediment, corals can also initiate an acute response to attempt to move the sediment using active mechanisms. Active sediment removal mechanisms include ciliary and tentacle movement, increased mucus production, and hydrostatic inflation (*Rogers, 1990*; *Stafford-Smith & Ormond, 1992*; *Stafford-Smith, 1993*; *Bessell-Browne et al., 2017*). However, these active mechanisms are often very energetically costly, and thus cannot be sustained for long periods of time (*Riegl & Branch, 1995*; *Erftemeijer et al., 2012*). If the sediment deposition rate exceeds the coral's sediment clearance rate, sediment will accumulate on the coral, reducing heterotrophic feeding and light transmission to algal endosymbionts and creating hypoxic conditions near the coral tissue, which often leads to tissue necrosis and coral mortality (*Philipp & Fabricius, 2003*; *Weber, Lott & Fabricius, 2006*; *Weber et al., 2012*).

Corals can also, or alternatively, interact with suspended sediment, which occurs when particles such as clay, silt, and sand are moved into the water column by some natural or anthropogenic disturbance and remain in the water column for a period of time (*Rogers, 1990*; *Fabricius, 2005*; *Erftemeijer et al., 2012*). Suspended sediment reduces the amount of light that reaches the coral, impeding the ability of the algal endosymbionts to photosynthesize and provide the coral host with sufficient energy for metabolism and growth (*Rogers, 1990*; *Fabricius, 2005*; *Erftemeijer et al., 2012*; *Bessell-Browne et al., 2017*). Reduced photosynthetic efficiency can induce corals to switch to heterotrophic feeding, a much less efficient way to obtain carbon than through its endosymbionts (*Muscatine & Porter, 1977*; *Anthony & Fabricius, 2000*; *Houlbrèque & Ferrier-Pagès, 2009*). Additionally, heterotrophic feeding in the presence of sediments may lead the coral to ingest sediment particles, disrupting its nutritional intake and potentially acting as a vector for harmful

bacteria and toxins (*Erftemeijer et al., 2012*; *Studivan et al., 2022*). Suspended sediment has also been observed to induce immune responses and increase disease prevalence in corals (*Pollock et al., 2014*; *Sheridan et al., 2014*).

In addition to sediment type, the morphology of the coral can modulate its interaction with sediments. For example, massive, plating, and encrusting corals have a higher planar surface area and thus higher capacity to trap sediments in comparison to branching corals with high three-dimensional and more vertical structure. Sediment removal from massive corals often requires active removal mechanisms (*Dallmeyer, Porter & Smith, 1982*; *Rogers, 1990*; *Stafford-Smith, 1993*). However, massive corals may also be more resilient to high suspended sediment concentrations because their greater surface area allows for increased opportunities to capture light, maximizing the photosynthetic efficiency of their algal endosymbionts (*Fabricius, 2005*; *Erftemeijer et al., 2012*). In contrast, the relatively small surface area and vertical branches of branching coral species means sediment is minimally trapped and can be more easily removed by gravity or currents (*Lasker, 1980*; *Rogers, 1983*; *Stafford-Smith, 1993*). Branching coral species typically have faster clearance rates than non-branching species, and active removal mechanisms are required less frequently, allowing branching corals to devote that energy towards other functions such as reproduction and growth (*Stafford-Smith, 1993*; *Jones, Fisher & Bessell-Browne, 2019*). Collectively, these studies support that morphology plays a role in response to sediment stress.

While most research has focused primarily on physiological responses to sediment stress, there is a small but growing body of work on gene expression of corals exposed to sediment stress. Early microarray studies found that the upregulation of heat shock proteins (HSPs) occurred in response to sediment stress (*Wiens et al., 2000*; *Hashimoto et al., 2004*). As a generalized stress response protein, HSPs have also been implicated in coral response to other stressors, such as deoxygenation, temperature and ocean acidification (*DeSalvo et al., 2008*; *DeSalvo et al., 2010*; *Kaniewska et al., 2012*; *Alderdice et al., 2022*). Another general response to sediment stress is upregulation of biomarkers of oxidative stress (*Morgan, Edge & Snell, 2005*), more specifically, thioredoxin, a protein that modulates redox and cell-to-cell signaling (*Tomanek, 2015*). A potentially more specific gene responding to sediment stress is indicated by the differential expression of urokinase plasminogen activator surface receptor (uPAR) transcripts in the coral *Diploria strigosa* along a sedimentation/pollution gradient in Castle Harbor, Bermuda (*Morgan, Edge & Snell, 2005*). uPAR is associated with proteolysis, wound healing and inflammation, and is hypothesized to contribute to coral tissue remodeling in response to elevated levels of sedimentation (*Morgan, Edge & Snell, 2005*). Genes related to immunity, as well as energy metabolism, were also implicated in the transcriptomic response to sediment stress in corals from two locations, Singapore (*Goniastrea pectinata* and *Mycedium elephantotus*) and Eilat, Israel (*G. pectinata* only) using RNASeq (*Bollati et al., 2021*). While there were some methodological differences between their experiments, shared mechanisms were identified across different species and populations, demonstrating a conserved response to sediment stress across species and sediment types. However, the corals evaluated in that study, *G. pectinata* and *M. elephantotus*, both have similar morphological characteristics

(massive and encrusting), indicating that this response may only be relevant to massive and encrusting corals (*Bollati et al., 2021*).

While these initial molecular analyses have provided important insights into the coral transcriptomic response to sediment stress, less is known about shared molecular responses by morphology and location/sediment type. To this end, our study aims to fill these knowledge gaps by examining gene expression across different coral morphologies and the use of multiple locations/types of sediment. Here we quantified the transcriptomic responses of corals with different colony morphologies in response to different types of sediment stress. Floridian corals (*Acropora cervicornis*, *Montastraea cavernosa* and *Orbicella faveolata*) were exposed to sterilized white carbonate sediment for 18 days, whereas Hawaiian corals (*Montipora capitata*, *Pocillopora acuta* (formerly *Pocillopora damicornis*) and *Porites lobata*) were exposed to unsterilized terrigenous red soil for up to 7 days. In this study, *A. cervicornis* and *P. acuta* were categorized as branching corals, while *M. cavernosa*, *O. faveolata* and *P. lobata* were categorized as massive corals. The morphology of *M. capitata* was considered as intermediate between branching and plating, as *M. capitata* tends to form plates growing horizontally with branches sprouting upward (*Veron, 2002*). The methodological differences prevent us from making direct statistical comparisons between the experiments. However, it is still relevant to highlight general biological processes and mechanisms related to sediment stress responses across morphology and location.

## METHODS

To characterize a broad range of responses to different kinds of sediment stress, two independent experiments were performed in Florida and Hawaiʻi; the separate experiments will be referred to as either Florida or Hawaiʻi throughout the text (Fig. 1). In the first experiment, Hawaiian corals were exposed to live terrigenous red soil for up to 7 days. In the other experiment, Floridian corals were exposed to sterilized coral rubble sediment for 18 days. We acknowledge that differences in experimental sediment regimes do not allow for direct statistical comparisons between these two experiments at all levels. However, qualitatively examining the patterns found in response to sediment across two experiments and in orthogroups across taxa in response to sediment provides the unique opportunity to describe these findings and test for shared transcriptomic patterns in response to sediment stress. Portions of this text were previously published as part of a preprint (*Ashey et al., 2023*; https://www.biorxiv.org/content/10.1101/2023.01.30. 526279v1.full).

### Collections and exposures

#### Hawaiʻi

All coral collections were obtained under Hawaiʻi SAP permit (SAP 2015-48). *Montipora capitata*, *Porites lobata* and *Pocillopora acuta* adult corals were collected in early June 2015 from Kāneʻohe Bay, Oʻahu, Hawaiʻi (21°25′59.1″N 157°47′11.1″W). One fragment was collected per colony, and fragments were acclimated to tank conditions for 1 week post collection. Following acclimation, 12 fragments per species were placed in six 60-L
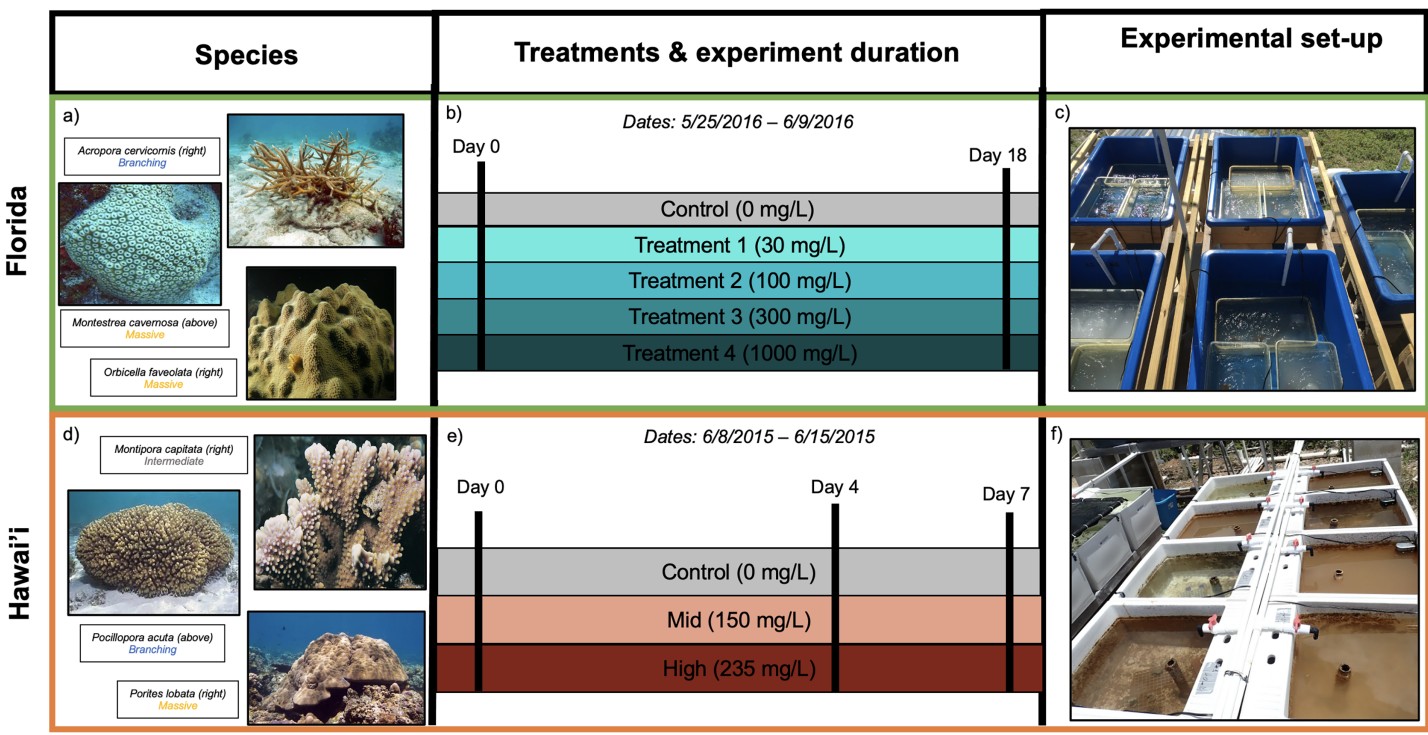

**Figure 1** **Experimental design for Florida (A–C) and Hawai'i (D–F) experiments.** (A) Species used and their morphologies in parentheses for Florida study: *Acropora cervicornis* (branching morphology), *Montastrea cavernosa* (massive morphology) and *Orbicella faveolata* (massive morphology). (B) Illustration of timeline and sediment treatments. (C) Experimental tanks from the Florida experiment (image by Francois Seneca). (D) Species used and their morphologies in parentheses for Hawai'i study: *Montipora capitata* (intermediate morphology), *Pocillopora acuta* (branching morphology) and *Porites lobata* (massive morphology) (E) Illustration of timeline and sediment treatments. (F) Experimental tanks from the Hawai'i experiment (image by Francois Seneca). Images of coral species were obtained from Corals of the World (*Veron et al., 2016a*, *2016b*, *2016c*, *2016d*, *2016e*, *2016f*) (C) Corals of the World.

flow-through outdoor tanks (two replicate tanks per treatment and two replicate fragments per tank for $n = 4$ samples per treatment, except *M. capitata* where $n = 3$) at the Point Lab of the Hawai'i Institute of Marine Biology (HIMB). Corals in Hawai'i have become increasingly exposed to sedimentation, agricultural runoff, and pollution that contains organic material (*Pastorok & Bilyard, 1985*; *Hunter & Evans, 1995*; *Ogston & Field, 2010*; *Weber et al., 2012*; *Abaya et al., 2018*), and the increased prevalence of organic matter in sediment and outflows can negatively affect corals (*Hodgson, 1990*; *Weber, Lott & Fabricius, 2006*; *Weber et al., 2012*; *Loiola, Oliveira & Kikuchi, 2013*). Given the prevalence of organic matter in sediment in Hawai'i, fragments were exposed to sediment originating from a live terrigenous Hawaiian red soil collected from the highest elevation point on Moku O Lo'e (Coconut Island). The total suspended sediment (TSS) concentration was determined by the technique described in *Cortés & Risk (1985)*. The unfiltered unsterilized terrigenous red dirt was mixed with fresh water to mimic the sediment moved during a storm and extract the silt and clay (<60 micron) used in the experiment as in *Bahr et al. (2020)*. Particles were filtered through mesh sieves to remove debris and produce the final silt-clay mixture. Sediment was added from a concentrated stock at day 0 and again at day 4 to reach a low and a high total suspended sediment (TSS) concentration, immediately

following addition. To test the efficacy of our additions, triplicate samples were taken from the experimental tanks following sediment dosing to measure the sediment load. The sediment samples were filtered and oven dried on Whatman paper, and the sediment weight measured with a precision scale at 150 ± 73 and 235 ± 63 mg/L, respectively (*Philipp & Fabricius, 2003*; *Browne et al., 2014*). Powerheads (Eheim Universal 300 L/h) were placed at the bottom of each tank to ensure water motion. Seawater was pumped directly from the lagoon in front of the Point Lab and was filtered through a sand filter to remove sediment before being distributed to the experiment tanks at a flow-through rate of 12 L/h. The temperature exposure was the natural profile of the lagoon water by HIMB during the time of the experiment (Station 51207; NOAA Buoy in Kāneʻohe Bay, Hawaiʻi; Fig. S1A). Salinity and total pH fluctuated between 34–35 psu and 7.90–8.01 during the experimental period, following the water conditions of the lagoon (MPCO2 Mooring Kāneʻohe, NOAA Buoy). The irradiance within the 60-liter experimental tanks was reduced by 30% with shade cloth to obtain a maximal daylight value fluctuating approximately between 60–90 lux, measured with a HOBO pendant data logger (HOBO Pendant Temp/Light 64 K logger) placed at the bottom of the tanks. Tanks received natural day/night light cycles. The coral fragments were not fed in this experiment. Coral fragments ($n = 4$ replicate fragments per species), two from each of the two tanks, in each of the three treatments were exposed (except *M. capitata* where $n = 3$) and were collected and preserved in liquid nitrogen at day 4 (24 samples on June 12, 2015), and day 7 (24 samples on June 15, 2015) of the exposure. Samples were then stored at −80 °C until extraction.

### Florida

Adult *Acropora cervicornis*, *Montastraea cavernosa*, and *Orbicella faveolata* colonies were collected from the Key West nursery of the Florida Keys National Marine Sanctuary on March 23, 2016 under NOAA National Marine Sanctuaries Permit #FKNMS-2015-016 and #FKNMS-2016-017. On March 28, 2016, coral colonies were transported from Key West to Fort Pierce in bins filled with aerated seawater and placed in acclimatation tanks at the Smithsonian Marine Station (Fort Pierce, FL, USA) prior to the start of the experiment. After acclimation, 15 fragments per species were placed in fifteen 12-L static outdoor tanks (three replicate tanks per treatment and 1 fragment per tank for $n = 3$ samples per treatment). The water source was oceanic water and had a consistent salinity of 35 psu. The experimental tanks were static, and reverse osmosis water was added daily to maintain the salinity. Tanks were submerged in recirculating water baths to keep them at a constant temperature of 28 °C. pH was not recorded during the experiment. Corals in Florida are subject to an increased number of sedimentation events, in particular through dredging and coastal development (*Barnes et al., 2015*; *Miller et al., 2016*; *Cunning et al., 2019*); these activities disturb bottom/seafloor sediments, which reduces light availability and can smother corals, leading to a multitude of adverse effects (*Erftemeijer et al., 2012*; *Jones et al., 2016*). Given the increased prevalence of dredging and coastal development in Florida, coral fragments were exposed to sterilized carbonate sand sediment intended to mimic dredging and development activities. Sediment was collected from Key Largo

(25°08′22.0″N 80°23′37.6″W) and was filtered through a 63 micron sieve to remove debris and obtain a fine grain mixture. Sediment was then run through active carbon filters for a week in seawater to reduce chemical pollution. Sterilization of the sediment stock was performed using an autoclave program for liquid (20 min at 121 °C and 2.1 bar). Despite being sterilized, the sediment still had the capacity to induce adverse effects in corals. Previous studies have found that sterilized sediment can deplete energy reserves, induce immune responses, and decrease photosynthetic capabilities (*Browne et al., 2014*; *Junjie et al., 2014*; *Sheridan et al., 2014*). Thus, it is likely that the sterilized sediment stressor used in the Florida experiment induced an ecologically relevant response. The total suspended sediment (TSS) concentration was determined by the technique described in *Bahr et al. (2020)*. On the first day of exposure (May 23, 2016), sediment was added from the sterilized concentrated stock (458 g/L). Sediment was added again on May 25, 27, and 30 to mimic repeat dredging activities. There were five total suspended sediment (TSS) 0, 30, 100, 300, 1,000 mg/L targets (*Rogers, 1979*; *Junjie et al., 2014*). To approximately obtain these TSS concentrations, 1, 3, 10, and 30 mL of stock was added to each 12-L treatment tank. Powerheads (Eheim Universal 300 L/h) were placed at the bottom of each tank to ensure water motion. Tanks received natural day/night cycles and were under 50% reduction shade, measured with a HOBO pendant data logger (HOBO Pendant Temp/Light 64 K logger) placed at the bottom of the tanks. The coral fragments were not fed in this experiment. On June 9, coral fragments from all tanks were collected and frozen in liquid nitrogen. Samples were then stored at −80 °C until extraction.

## RNA extraction

Prior to RNA extraction, frozen coral samples were crushed with a manual hydraulic press (12 tons pressure) and a metal mortar and pestle chilled with liquid nitrogen. Approximately 100 mg of frozen coral powder was used in the RNA extractions starting with 1 mL of TRIzol™ Reagent (Invitrogen, Waltham, MA, USA) and 100 uL of 0.1 mm ceramic beads in 2.0 mL tubes. Coral tissues were lysed with two rounds of 20 s at 6,500 bpm on a tissue homogenizer (FastPrep), and a rest on ice of 30 s in between rounds. Tissue slurries were then incubated on ice for 5 min. After a brief spin down to gather liquid at the bottom of the tube, 300 uL of molecular grade chloroform was added to the slurry. Tubes were hand shaken for 15 s and rested for 3 min on ice. Phase separation was obtained by centrifugation at 12 xg for 15 min at 4 °C. Top aqueous phase containing RNA was transferred to new 1.5 mL tubes and mixed again with 200 uL of chloroform. The previous three steps were repeated. An equal volume of chilled 70% molecular grade ethanol was added to the RNA in solution. From this point, the Direct-zol™ RNA MiniPrep (Zymo Research; Cat# R2070, Orange, CA, USA) protocol with DNase treatment was followed. Total RNAs were eluted in DEPC-treated water and quantified using the Qubit fluorometer. RNA quality was assessed using a NanoDrop spectrophotometer in addition to a visual check for degradation *via* a 1% agarose TAE gel. Samples with at least 9 ug/uL of RNA were sent for sequencing.

## RNA sequencing

RNA samples were diluted to accommodate for library production starting with 100 ng of total RNA. Samples were then loaded onto Neoprep cards and processed following the TruSeq stranded mRNA Library Prep for NeoPrep kit (Document # 15049725 v03; Illumina, San Diego, CA, USA) protocol. Quality controlled libraries were then sequenced through HiSeq 50 cycle single read sequencing v4 by the High Throughput Genomics Core Facility at the University of Utah.

## Bioinformatic analysis

Workflows and data are located at https://github.com/JillAshey/SedimentStress. Quality checks of raw and trimmed reads were performed using FASTQC (v0.11.8, Java-1.8; *Andrews, 2010*) and MultiQC (v1.7, Python-2.7.15; *Ewels et al., 2016*). Reads that did not pass quality control were trimmed with Trimmomatic (v0.30, Java-1.8; *Bolger, Lohse & Usadel, 2014*). Genomes, protein sequences, and transcript sequences were obtained from the following locations: *Acropora cervicornis* (Baums Lab, v1.0_171209; https://usegalaxy.org/u/skitch/h/acervicornis-genome); *Montastraea cavernosa* (Matz Lab, July 2018 version; https://matzlab.weebly.com/data–code.html); *Montipora capitata* (http://cyanophora.rutgers.edu/montipora/; Version 2, *Stephens et al., 2021*); *Orbicella faveolata* (NCBI, assembly accession GCF_002042975.1; *Prada et al., 2016*); *Pocillopora acuta* (http://cyanophora.rutgers.edu/Pocillopora_acuta/; Version 1, (*Stephens et al., 2021*); *Porites lutea* (used to analyze *P. lobata* data in current study; http://plut.reefgenomics.org/download/; Version 1.1, *Robbins et al., 2019*). We have archived all references used for this analysis at https://osf.io/8qn6c/ (DOI 10.17605/OSF.IO/8QN6C) to enable reproducible analyses for this project. After trimming, reads were mapped to their respective genomes using STAR (v2.5.3; *Dobin et al., 2013*). The aligned read files from STAR (BAM file format) were assembled to the references and count data were generated using StringTie (v2.1.1-GCCcore-7.3.0; *Pertea et al., 2015*). Assembly quality was assessed with gffcompare (v0.11.5; *Pertea & Pertea, 2020*). The StringTie *prepDE* python (v2.7.15; *Pertea et al., 2015*) script was used to generate a gene count matrix.

To generate current gene ontology information for all species, functional annotation was performed on all genomes using the following workflow (https://github.com/JillAshey/FunctionalAnnotation). First, protein sequences from each species were identified using BLAST (*blastp*; v2.11.0; *Altschul et al., 1990*) against NCBI's nr database (1e-5 e-value; database accessed and updated on Oct. 12, 2021) and the Swissport database (1e-5 e-value; database accessed and updated on Oct. 22, 2021; *Bairoch & Apweiler, 1997*). The XML files generated from the BLAST output were then used as input for BLAST2GO (v.5.2.5) to generate gene ontology (GO) terms (*Götz et al., 2008*). Protein sequences were also used as input to InterProScan (v5.46–81.0; Java v11.0.2), which identified homologous sequences and assigned GO terms (*Jones et al., 2014*). Using BLAST2GO, the XML file generated from InterProScan was merged with the nr and Swissprot BLAST2GO output, creating final functional annotation tables, which were saved in csv format (https://osf.io/8qn6c/).

## Gene expression and ontology analysis

All gene expression and ontology analyses were done in RStudio (v1.3.959) using v4.0.2 of R. First, genes were filtered using genefilter's (v1.70.0; *Gentleman et al., 2021*) *pOverA* function; genes were retained for expression analysis only if counts were greater than or equal to 5 in at least 85% of the samples, which minimizes differential expression results from low count genes with lower confidence. Because different samples may have been sequenced to different depths, size factors were calculated as the standard median ratio of a sample over a 'pseudosample' (for each gene, the geometric mean of all samples; *Anders & Huber, 2010*; *Love, Huber & Anders, 2014*). After confirming size factors were estimated to be less than 4, filtered gene counts were normalized using DESeq's (v1.28.1) *vst* function (*Love, Huber & Anders, 2014*). Treatment was set as a factor; in the HI experiment, Time (Day 4 and Day 7) was not significant as a factor and so corals sampled on different days were combined by treatment for further analysis. Differential gene expression was assessed using the *DESeq* function with the Wald likelihood test ratio ($p$-adjusted $< 0.05$; *Love, Huber & Anders, 2014*). PCA plots with differentially expressed genes were generated using the *plotPCA* function (*Love, Huber & Anders, 2014*). To estimate the power to detect a range of effect sizes (1.25, 1.5, 1.75, and 2), the R package RNASeqPower (v1.38.0; *Hart et al., 2013*). Depth of sequencing was calculated using the formula derived from Sims et al. (2014) as LN/G, where L is read length, N is the average number of reads and G is the transcriptome length. The power analysis using the filtered gene datasets (Table S1) indicates the power to detect an effect was similar across the six species (effects size standard error of the mean ranged from 0.012 to 0.06 across the range of effect sizes tested).

Gene ontology analysis was completed with GOSeq (v1.40.0; *Young et al., 2010*), which corrects for the higher-confidence in differential expression as a function of gene length as follows: Genes that passed the *pOverA* filter and were marked as differentially expressed genes by DESeq above were used to calculate the probability weighting function using function *nullp* with the bias data being gene length (*Young et al., 2010*). To identify category enrichment amongst differentially expressed genes, the *goseq* function was performed with the Wald method (*Young et al., 2010*). Significantly enriched GO terms from each of the biological processes (BP), molecular functions (MF), or cellular components (CC) ontologies were denoted as those with an over-represented $p$-value $< 0.05$. BP GO term analysis is presented here; CC and MF GO term analyses can be found in the Supplemental Material (See Figs. S2–S7). GO term information was organized under their parent GO slim terms (obtained from http://www.informatics.jax.org/gotools/data/input/map2MGIslim.txt; accessed on April 4, 2021) for qualitative comparison across species. Overlap of GO terms between species was compared and visualized using ComplexUpset's (v1.3.3; *Lex et al., 2014*) *upset* function.

## Orthology analysis

To test for functional similarities in response to sediment stress across taxa, all species were characterized into orthogroups using OrthoFinder (v2.3.3) with dependencies Diamond (v0.9.22), MCL (v14.137), FastME (v2.1.6.1) and BLAST+ (v2.8.1) and default parameters

**Table 1 Summary of read sequencing, quality and alignment by species.** See Table S1 for more details.

| Species | Location | # of samples | Avg. raw reads | Avg. clean reads | Avg. # of reads removed | Avg. % of reads mapped |
|---------|----------|--------------|----------------|-------------------|--------------------------|-------------------------|
| *A. cervicornis* | Florida | 13 | 16,630,621 | 16,499,909 | 130,712 | 70.62 |
| *M. cavernosa* | Florida | 15 | 15,921,959 | 15,740,195 | 181,765 | 50.18 |
| *O. faveolata* | Florida | 14 | 14,111,857 | 13,905,076 | 206,781 | 65.54 |
| *M. capitata* | Hawai'i | 11 | 16,360,670 | 16,280,212 | 80,459 | 63.75 |
| *P. acuta* | Hawai'i | 12 | 14,441,259 | 14,359,654 | 81,605 | 68.94 |
| *P. lobata* | Hawai'i | 12 | 13,995,393 | 13,858,883 | 136,510 | 52.83 |

(*Emms & Kelly, 2015*, *2019*). Orthogroups were filtered to those common in all species. The differentially expressed genes (DEGs) in the orthogroups for each species were identified using the DESeq2 results. Commonality in functional orthogroups containing DEGs were compared between species and visualized using ComplexUpset's (v1.3.3; *Lex et al., 2014*) *upset* function.

# RESULTS

## Read sequencing and quality

Sequencing of 77 samples ($n$ = 11–15 per species; Table 1) yielded a total of 1,173,800,658 raw reads with an average of 15,244,164 ± 3,285,387 raw reads per sample (Table S2). Quality filtering and trimming removed an average of 140,560 ± 119,359 reads per sample, leaving an average of 15,103,605 ± 3,290,800 cleaned reads per sample for analysis (Table S2). Reads were aligned to the species-specific genome, and alignment rates ranged from an average of 50.18% in *M. cavernosa* to an average of 70.62% in *A. cervicornis* (Table 1; Table S2).

## Gene expression

For Florida species, *A. cervicornis* had 215 unique differentially expressed genes (DEGs), *M. cavernosa* had 62 DEGs, and *O. faveolata* had 8 DEGs between ambient and sediment exposure treatments (Table 2; Table S3). Among these, 67 genes in *A. cervicornis*, 19 in *M. cavernosa*, and 0 in *O. faveolata* were upregulated (Log Fold Change, LFC > 0, padj < 0.05), while 154 genes in *A. cervicornis*, 44 in *M. cavernosa*, and 8 in *O. faveolata* were downregulated (LFC < 0, padj < 0.05; Table 2; Table S3). In PCA plots of all genes for Hawaiian species, 'Days' was not visually separated from 'Treatment'; therefore, 'Days' was dropped as an independent variable in further analyses. For Hawaiian coral species, when comparing ambient and sediment exposure treatments, *M. capitata* had 157, *P. acuta* had 263, and *P. lobata* has 153 unique DEGs (Table 2; Table S3). 79 genes in *M. capitata*, 129 in *P. acuta* and 32 in *P. lobata* were upregulated (LFC > 0, padj < 0.05), while 78 genes in *M. capitata*, 135 in *P. acuta* and 128 in *P. lobata* were downregulated (LFC < 0, padj < 0.05; Table 2; Table S3). Interestingly, there were more downregulated DEGs than upregulated in all species examined, with the exception of *M. capitata*. It is possible that the lower numbers of DEGs in the Florida coral group was due, in part, to a level of acclimation to

**Table 2 Summary of differentially expressed genes (DEGs), gene ontology (GO) terms and orthogroups by species.** See Tables S2–S5 for more details.

| Species | Loca-tion | Morpholo-gy | Unique DEGs | Up-regulated DEGs | Down-regulated DEGs | Unique GO terms assigned to DEGs | Orthogroups containing DEGs | DEGs in orthog-roups |
|---|---|---|---|---|---|---|---|---|
| *A.cervicornis* | Florida | Branching | 215 | 67 | 154 | 278 | 102 | 119 |
| *M. cavernosa* | Florida | Massive | 62 | 19 | 44 | 158 | 28 | 31 |
| *O. faveolata* | Florida | Massive | 8 | 0 | 8 | 27 | 2 | 2 |
| *M. capitata* | Hawai'i | Intermediate | 157 | 79 | 78 | 237 | 80 | 87 |
| *P. acuta* | Hawai'i | Branching | 263 | 129 | 135 | 380 | 100 | 123 |
| *P. lobata* | Hawai'i | Massive | 153 | 32 | 128 | 198 | 56 | 66 |

the sediment stress, as the Florida experiment was longer (18 days) than the Hawai'i experiment (up to 7 days).

PCA of the DEGs for each species showed that all sediment treatments clustered away from the control on the PC1 axis (PC1 variance explained was 51% for *A. cervicornis*, 46% for *M. cavernosa*, 51% for *O. faveolata*, 79% for *M. capitata*, 64% for *P. acuta*, and 50% for *P. lobata*), supporting a concerted gene expression response to sediment stress across taxa (Figs. 2A–2F). There was slight separation between the mid and high treatments on the PC2 axis for both *P. acuta* and *P. lobata*, indicating differential responses to sediment concentration (PC2 variance explained was 13% for *P. acuta* and 22% for *P. lobata*; Figs. 2E and 2F).

## Gene ontology

Significantly enriched gene ontology (GO) terms were identified in the DEGs of the species in the present study. In the Florida experiment, corals exposed to sterilized carbonate sediment for 18 days resulted in 278 unique GO terms assigned to *A. cervicornis* DEGs, 158 to *M. cavernosa* DEGs, and 27 to *O. faveolata* DEGs (Table 2; Table S2). *A. cervicornis* shared GO terms related to developmental processes and signal transduction with both *M. cavernosa* and *O. faveolata* (Table 3; Fig. 3; Table S4). Specifically, ovarian cumulus expansion (GO:0001550), positive regulation of skeletal muscle tissue development (GO:0048643) and regulation of Rho protein signal transduction (GO:0035023) were shared between *A. cervicornis* and *M. cavernosa*, while chondrocyte development (GO:0002063) and positive regulation of signal transduction (GO:0009967) were shared between *A. cervicornis* and *O. faveolata* (Table 3; Fig. 3; Table S4). Despite similar morphologies, *M. cavernosa* and *O. faveolata* did not share any GO terms.

In the Hawaiian coral species that were exposed to unsterilized red terrigenous sediment for up to 7 days, 237, 380, and 198 GO terms were assigned to *M. capitata*, *P. acuta*, and *P. lobata* DEGs, respectively (Table 2; Table S2). Grouping by GO slim term, we found that GO enrichment occurred in functions related to developmental processes, protein metabolism, cell organization and biogenesis, signal transduction, and stress response, among others (Table 3; Fig. 3; Table S4). Only one shared GO term was found in all three
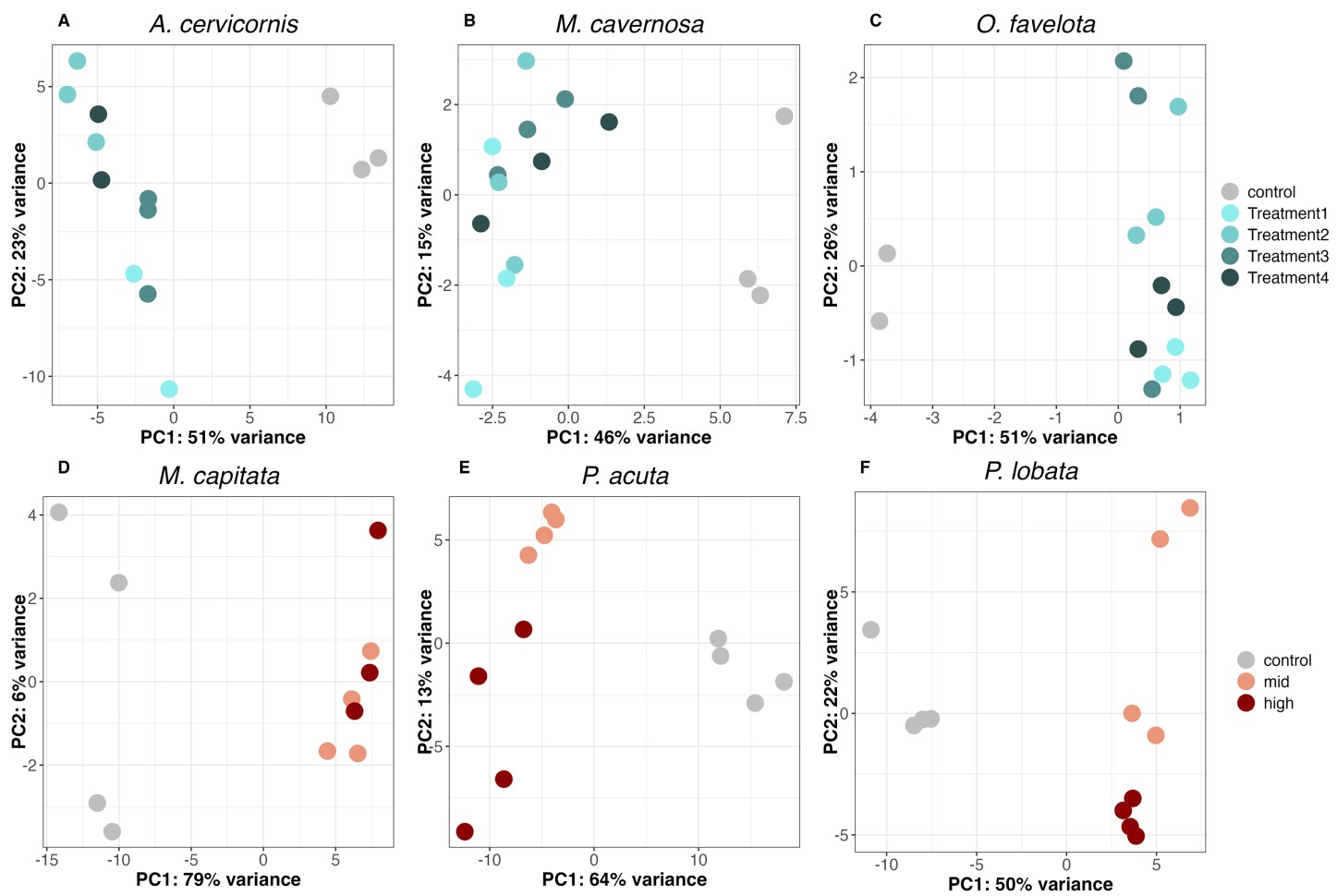

**Figure 2 Principal component analysis (PCA) of differentially expressed genes in (A)** *A. cervicornis*, **(B)** *M. cavernosa*, **(C)** *O. faveolata*, **(D)** *M. capitata*, **(E)** *P. acuta*, **and (F)** *P. lobata*.

of *M. capitata*, *P. acuta*, and *P. lobata*, namely microtubule-based processes (GO:0007017; Table 3; Fig. 3; Table S4). *M. capitata* and *P. acuta*, both of which were branching corals, shared 20 GO terms, primarily relating to developmental processes and signal transduction (Table 3; Fig. 3; Table S4). This was the largest number of GO terms shared between species. *P. acuta* and *P. lobata* shared functional responses related to cell organization and biogenesis, cell-cell signaling, and developmental processes. Developmental processes included five shared GO terms, specifically embryonic axis specifications (GO:0000578), embryonic organ development (GO:0048568), eye development (GO:0001654), neural crest cell fate specification (GO:0014036), and neural plate anterior/posterior regionalization (GO:0021999). *P. acuta* had the most GO terms identified across both experiments (Table 3; Fig. 3; Table S4). Aside from the microtubule-based process, *M. capitata* and *P. lobata* did not share other GO terms.

There was overlap of GO terms between location and morphology (Table 3; Figs. 4 and 5). *M. capitata* was present in all in common three-way interactions; it shared positive regulation of skeletal muscle tissue development (GO:0048643) with *M. cavernosa* and

**Table 3  Shared Biological Processes gene ontology (GO) terms between species.** See Tables S2 & S3 for more details.

| Species | # of shared GO terms | GO slim category | GO term |
|---|---|---|---|
| *A. cervicornis* + *M. cavernosa* + *M. capitata* | 1 | Developmental processes | Positive regulation of skeletal muscle tissue development (GO: 0048643) |
| *M. capitata* + *M. cavernosa* + *P. acuta* | 1 | Transport | Riboflavin transport (GO:0032218) |
| *M. capitata* + *P. acuta* + *P. lobata* | 1 | Other biological processes | Microtubule-based process (GO:0007017) |
| *A. cervicornis* + *M. cavernosa* | | Cell-cell signaling & transport | Regulation of neurotransmitter uptake (GO:0051580) |
| | | Developmental processes | Ovarian cumulus expansion (GO:0001550) |
| | | Other metabolic processes | Diadenosine tetraphosphate biosynthetic process (GO:0015966) |
| | | Signal transduction | Regulation of Rho protein signal transduction (GO:0035023) |
| | | No GO slim category identified | Negative regulation of chondrocyte proliferation (GO:1902731) |
| | 11 | | Positive regulation of cell junction assembly (GO:1901890) |
| | | | Negative regulation of histone H4-K16 acetylation (GO:2000619) |
| | | | Plasma membrane raft assembly (GO:0044854) |
| | | | Cellular response to exogenous dsRNA (GO:0071360) |
| | | | Cellular response to platelet-derived growth factor stimulus (GO: 0036120) |
| | | | Positive regulation of extrinsic apoptotic signaling pathway *via* death domain receptors (GO:1902043) |
| *A. cervicornis* + *O. faveolata* | 2 | Developmental processes | Chondrocyte development (GO:0002063) |
| | | Signal transduction | Positive regulation of signal transduction (GO:0009967) |
| *A. cervicornis* + *M. capitata* | 1 | No GO slim category identified | positive regulation of adipose tissue development (GO:1904179) |
| *A. cervicornis* + *P. acuta* | 5 | Other biological processes | Maintenance of gastrointestinal epithelium (GO:0030277) |
| | | Signal transduction | Wnt receptor signaling pathway (GO:0016055) |
| | | Protein metabolism | Mo-molybdopterin cofactor biosynthetic process (GO:0006777) |
| | | Stress response | Cellular response to starvation (GO:0009267) |
| | | No GO slim category identified | Mesenchymal stem cell maintenance involved in nephron morphogenesis (GO:0072038) |
| *A. cervicornis* + *P. lobata* | 3 | RNA metabolism | Positive regulation of transcription by RNA polymerase II (GO: 0045944) |
| | | | DNA methylation-dependent heterochromatin assembly (GO: 0006346) |
| | | Cell organization and biogenesis | DNA methylation-dependent heterochromatin assembly (GO: 0006346) |
| *M. capitata* + *M. cavernosa* | 4 | Cell organization and biogenesis | Positive regulation of neuron projection development (GO: 0010976) |
| | | Developmental processes | Positive regulation of neuron projection development (GO: 0010976) |
| | | | Retina development in camera-type eye (GO:0060041) |
| | | Other biological processes | Negative regulation of mitochondrial membrane potential (GO: 0010917) |
| | | No GO slim category identified | Ventricular compact myocardium morphogenesis (GO:0003223) |

(Continued)

| Table 3 (continued) | | | |
|---|---|---|---|
| Species | # of shared GO terms | GO slim category | GO term |
| *M. capitata* + *P. acuta* | 20 | Cell adhesion | Heterophilic cell adhesion (GO:0007157) |
| | | Developmental processes | Embryonic hindlimb morphogenesis (GO:0035116) |
| | | | Positive regulation of osteoclast differentiation (GO:0045672) |
| | | Other biological processes | Positive regulation of bone resorption (GO:0045780) |
| | | | Positive regulation of cellular pH reduction (GO:0032849) |
| | | | Oxidation reduction (GO:0055114) |
| | | | Response to activity (GO:0014823) |
| | | | Response to glucagon stimulus (GO:0033762) |
| | | | Response to pH (GO:0009268) |
| | | RNA metabolism | Positive regulation of transcription factor activity (GO:0051091) |
| | | Signal transduction | Blue light signaling pathway (GO:0009785) |
| | | Stress response | Response to pain (GO:0048265) |
| | | Transport | Carbon dioxide transport (GO:0015670) |
| | | | Secretion (GO:0046903) |
| | | No GO slim category identified | Angiotensin-activated signaling pathway (GO:0038166) |
| | | | Cellular response to retinoic acid (GO:0071300) |
| | | | Positive regulation of dipeptide transmembrane transport (GO:2001150) |
| | | | Positive regulation of mitochondrial membrane permeability (GO:0035794) |
| | | | Negative regulation of glucocorticoid secretion (GO:2000850) |
| | | | Regulation of chloride transport (GO:2001225) |
| *M. cavernosa* + *P. acuta* | 2 | Developmental processes | Ectoderm development (GO:0007398) |
| | | No GO slim category identified | Anterior head development (GO:0097065) |
| *M. cavernosa* + *P. lobata* | 3 | Developmental processes | Somatic muscle development (GO:0007525) |
| | | No GO slim category identified | Regulation of cell junction assembly (GO:1901888) |
| | | | Regulation of protein serine/threonine kinase activity (GO:0071900) |
| *O. faveolata* + *P. acuta* | 1 | Other biological processes | Response to stimulus (GO:0050896) |
| *P. acuta* + *P. lobata* | 11 | Cell-cell signaling | SPEMANN organizer formation (GO:0060061) |
| | | Cell organization and biogenesis | Neural crest cell fate specification (GO:0014036) |
| | | Developmental processes | Embryonic axis specification (GO:0000578) |
| | | | Embryonic organ development (GO:0048568) |
| | | | Eye development (GO:0001654) |
| | | | Neural crest cell fate specification (GO:0014036) |
| | | | Neural plate anterior/posterior regionalization (GO:0021999) |
| | | Other biological processes | Regulation of circadian rhythm (GO:0042752) |
| | | No GO slim category identified | Canonical Wnt signaling pathway involved in neural crest cell differentiation (GO:0044335) |
| | | | Cellular hypotonic response (GO:0071476) |
| | | | Endocardial cushion development (GO:0003197) |
| | | | Negative regulation of cardiac cell fate specification (GO:2000044) |

| Table 3 (continued) | | | |
|---|---|---|---|
| Species | # of shared GO terms | GO slim category | GO term |
| A. cervicornis | 256 | See Tables S2 and S3 | |
| M. capitata | 209 | See Tables S2 and S3 | |
| M. cavernosa | 136 | See Tables S2 and S3 | |
| O. faveolata | 24 | See Tables S2 and S3 | |
| P. acuta | 339 | See Tables S2 and S3 | |
| P. lobata | 181 | See Tables S2 and S3 | |

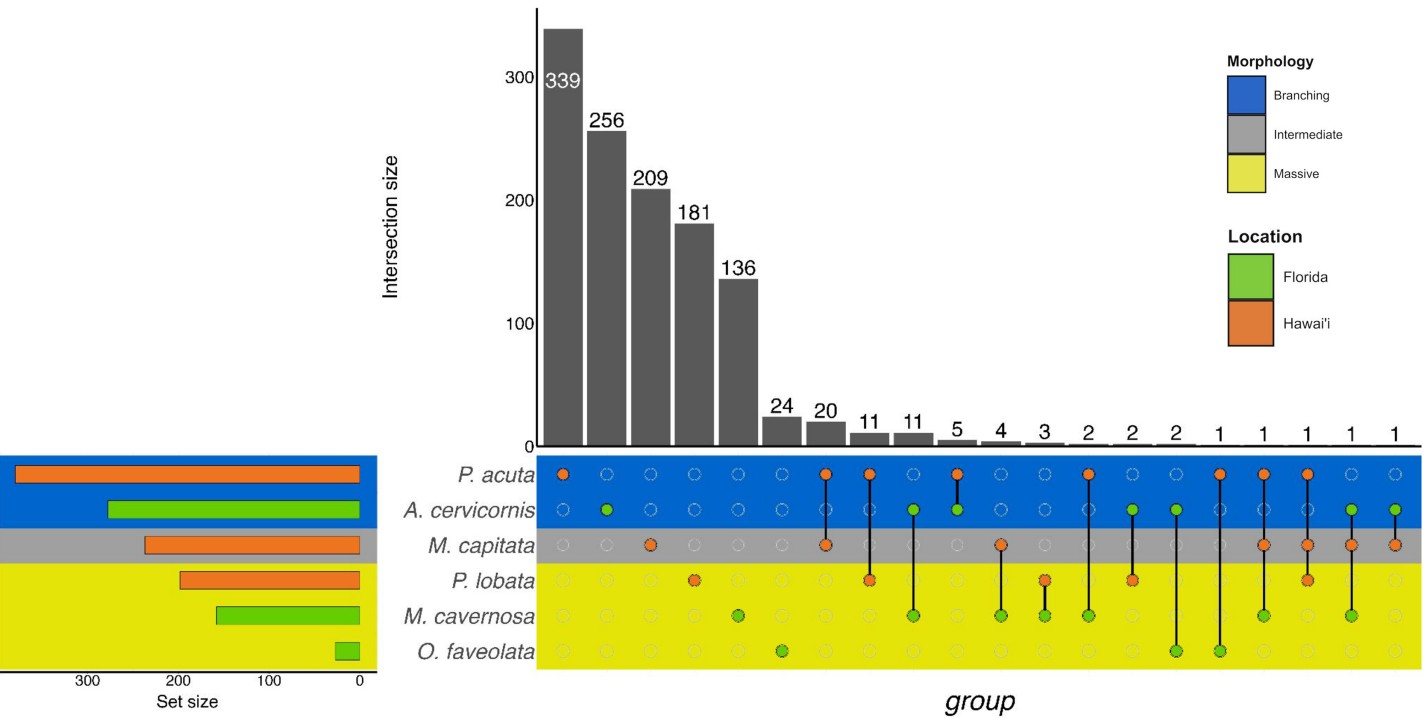

**Figure 3 UpSet plot of biological process GO terms across species.** UpSet plot of the intersection of biological process GO terms across location (green bar or circle = Florida; orange bar or circle = Hawai»i) and morphology (blue stripe = branching; gray stripe = intermediate; yellow stripe = massive).

*A. cervicornis*, riboflavin transport (GO:0032218) with *M. cavernosa* and *P. acuta*, and microtubule-based process (GO:0007017) with *P. acuta* and *P. lobata*. *M. capitata* and *M. cavernosa* also shared three GO terms related to developmental processes (positive regulation of neuron projection development (GO:0010976) and retina development in camera-type eye (GO:0060041; Table 3; Table S4). Among others, *M. cavernosa* from the FL experiment, shared GO terms relating to ectoderm development (GO:0007398) and regulation of protein serine/threonine kinase activity (GO:0071900) with *P. acuta* and *P. lobata*, respectively. *A. cervicornis* and *P. acuta*, both branching corals, shared 11 GO terms, including Mo-molybdopterin cofactor biosynthetic process (GO:0006777) and Wnt

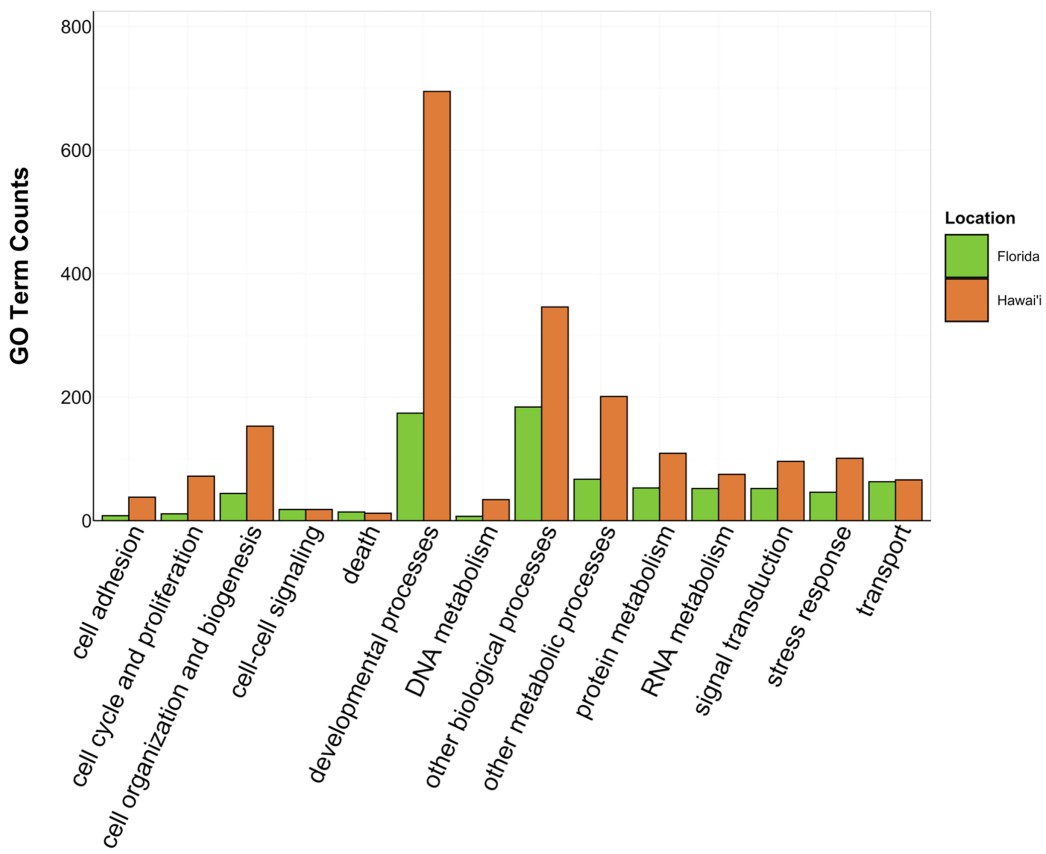

**Figure 4 Counts of biological process GO terms grouped under GO slim categories by location.** GO slim categories are on the x-axis, while the number of Biological Processes GO terms in each GO slim category is on the y-axis. The bars are colored by location: green bar = Florida, orange bar = Hawai'i.

receptor signaling pathway (GO:0016055). With the exception of *O. faveolata*, the massive corals, *M. cavernosa* and *P. lobata*, shared GO terms relating to regulation of cell junction assembly (GO:1901888), regulation of protein serine/threonine kinase activity (GO:0071900), and somatic muscle development (GO:0007525). In total, more significantly enriched GO terms were identified at HI and in the branching corals (Figs. 4 and 5).

## Orthogroups

In total, 21,688 total orthogroups were identified and 9,216 were common to all species. Amongst the Florida species, common orthogroups contained 119 DEGs from *A. cervicornis*, 31 from *M. cavernosa*, and two from *O. faveolata* (Table 2; Table S5). In the Hawai'i species, common orthogroups included 87 DEGs from *M. capitata*, 123 from *P. acuta*, and 66 from *P. lobata* (Table 2; Table S5).

Similarly to GO terms, there was overlap in orthogroups between species. There was one 4-way interaction, in which *A. cervicornis*, *M. capitata*, *P. acuta*, and *P. lobata* shared two orthogroups (Fig. 6; Table S5). *A. cervicornis* and *P. acuta* shared 3 orthogroups, the most orthogroups shared between a set of species (Fig. 6; Table S6). Although there may not

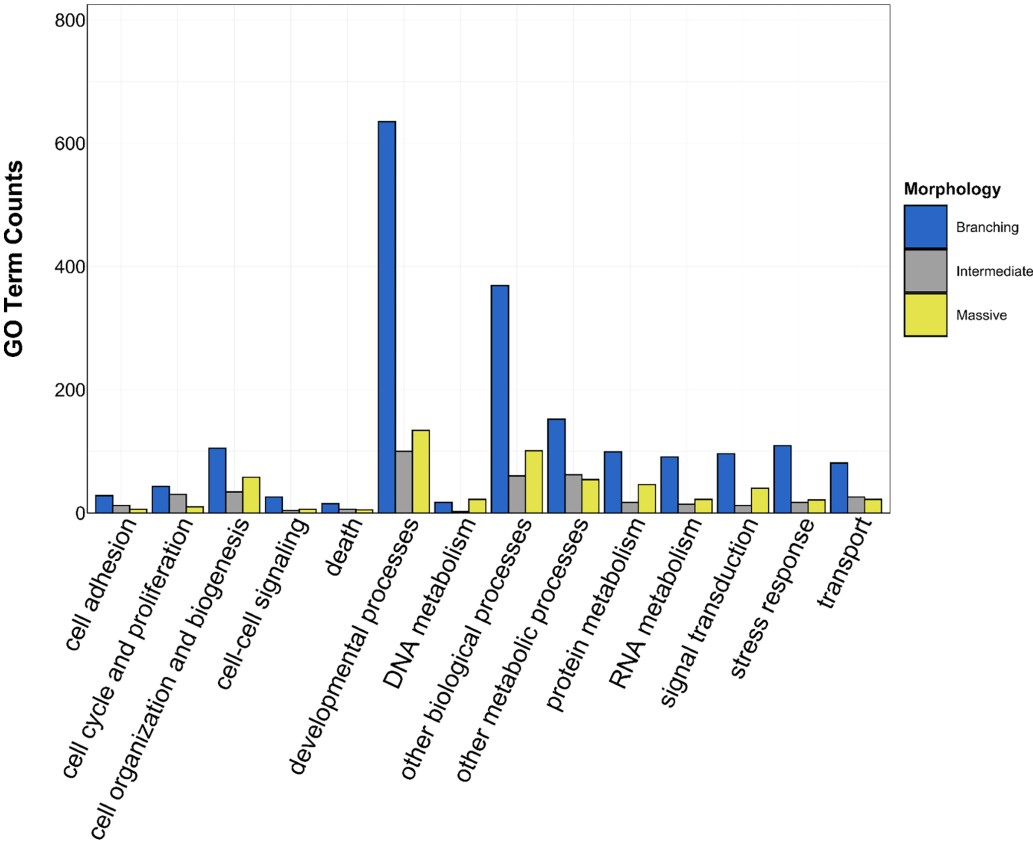

**Figure 5** **Counts of biological process GO terms grouped under GO slim categories by morphology.**
GO slim categories are on the x-axis, while the number of Biological Processes GO terms in each GO slim
category is on the y-axis. The bars are colored by morphology: blue bar = branching, gray bar = inter-
mediate, yellow bar = massive.    

have been a high number of overlapping orthogroups, the function of those overlapping
orthogroups were similar between species. For example, one orthogroup (OG0000487)
contained DEGs related to microtubule-based processes and structural constituent of
cytoskeleton in *M. capitata* and *P. acuta* (Table S6).

# DISCUSSION

In this study, we conducted two separate experiments to characterize the molecular
underpinnings of corals with differing morphological characteristics responding to
sediment stressors in two locations, Florida and Hawai'i. The methodological differences
prevent us from making direct statistical comparisons between the experiments. However,
it is still relevant to highlight general biological processes and mechanisms related to
sediment stress responses across morphology and location.

## Responses to unsterilized red sediment in Hawai'i

It is well established that morphology can play a role in modulating a coral's response to
sediment stress, but it is unknown if gene expression varies in corals with differing
morphological traits. Our study found that, across morphologies in Hawaiian corals,

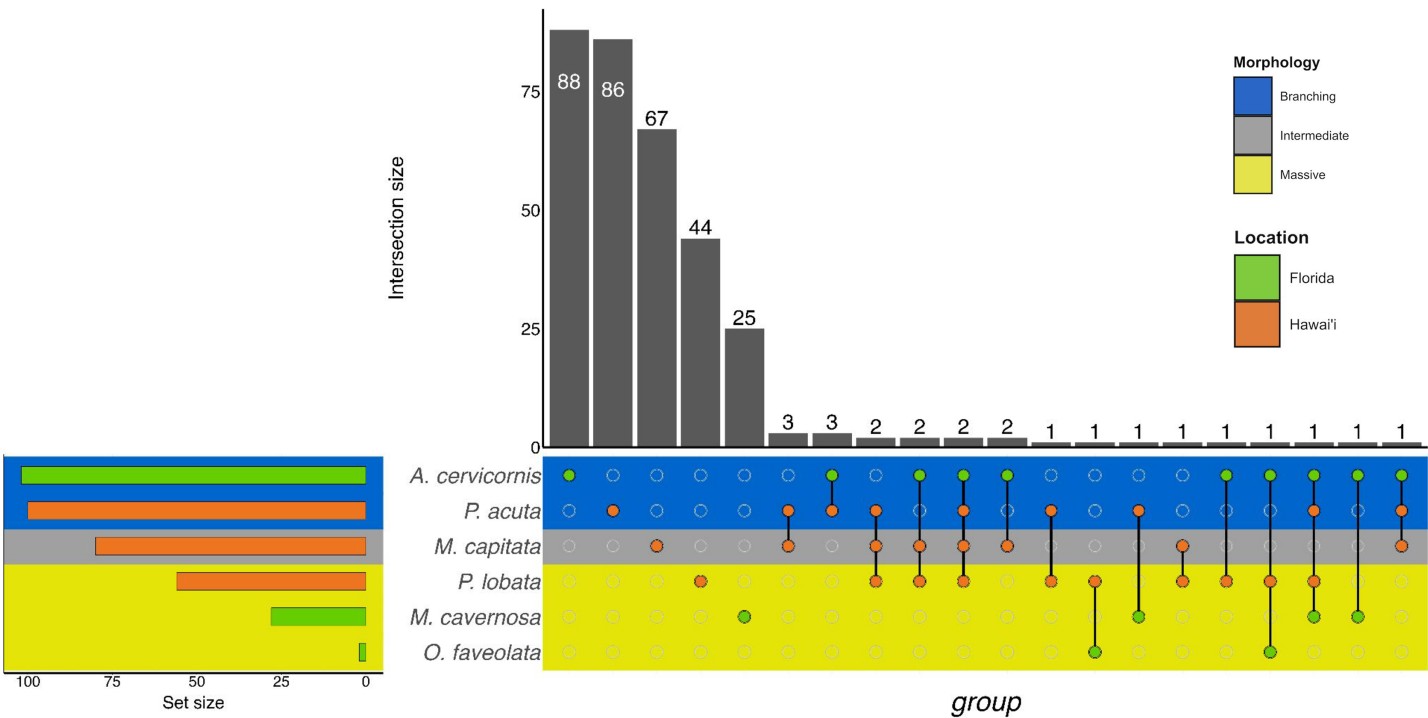

**Figure 6 UpSet plot of the intersection of orthogroups across location (green bar or circle = Florida; orange bar or circle = Hawaiʻi) and morphology (blue stripe = branching; gray stripe = intermediate; yellow stripe = massive).**

developmental processes were primarily affected by unsterilized sediment. Previous work has demonstrated that sediment deposition, high turbidity, and low light have been shown to adversely affect development and reproductive output across morphologies (*Kojis & Quinn, 1984*; *Gilmour, 1999*; *Humphrey et al., 2008*), though *M. capitata* development and fecundity are not always negatively affected by high sedimentation rates (*Padilla-Gamiño et al., 2014*). In the present study, the branching coral, *P. acuta*, shared a relatively high number of overlapping developmental processes-related responses with corals with intermediate (*M. capitata*; 20 shared terms) and massive (*P. lobata*; 11 shared terms) morphology. Shared responses corresponded to reproduction and developmental processes like embryonic axis specifications, embryonic organ development, eye development, neural crest cell fate specification, neural plate anterior/posterior regionalization, embryonic hindlimb morphogenesis and others. Unexpectedly, these terms primarily correspond to vertebrate developmental processes, complicating our ability to interpret these results. Given our use of specific annotation databases (*i.e.*, NCBI, SwissProt, InterPro), it is possible that these databases are all dominated by vertebrate-related annotations and thus, invertebrate protein sequences are assigned vertebrate-centric annotations. While it is clear that developmental processes are affected by short term sediment stress across morphologies, it is unclear how vertebrate specific gene functions may translate to developmental processes in basal metazoans. More work is necessary to evaluate equivalency across annotation softwares.

During the year, *M. capitata*, *P. acuta*, and *P. lobata* develop their gametes and then release when conditions are optimal, typically June and July in Hawai'i (*Stimson, 1978*; *Richmond & Jokiel, 1984*; *Padilla-Gamiño & Gates, 2012*; *Brown et al., 2020*). Because the corals in our study were exposed during this time period, it is possible that we captured gene expression signatures unique to corals at or near the peak of their reproductive phenotype. Although the corals were not sampled at different times of year under the same experimental conditions, these results suggest that because the corals were sampled during their reproductive period, signals of developmental processes may be higher than they might have been at other points in the year. Given the high number of shared and unique developmental process GO terms in the Hawai'i corals, however, it is likely that sediment has the potential to have reproductive and developmental consequences, which can have subsequent impacts on population growth and dynamics.

The only specific shared response across the three Hawaiian species was the downregulation of microtubule-based processes. Microtubules, tubulin polymers that maintain structure and shape to eukaryotic cells, are major components of cilia, which coral utilize for activities such as feeding and clearing sediment (*Westbroek, Yanagida & Isa, 1980*; *Rogers, 1990*; *Erftemeijer et al., 2012*). Downregulation of processes relating to cilia biogenesis/degradation and motility was also observed in the *P. acuta* host and its endosymbionts in response to combined acute heat and sediment stress (*Poquita-Du et al., 2019*, *2020*). The authors of these studies hypothesized that the corals were likely diverting energy resources away from feeding and active sediment clearing in order to prioritize energy for basic homeostasis. While our experiment did not include heat stress, it is probable that downregulation of microtubule or cilia-related processes is a generally conserved response to sediment stress. Exhaustion of sediment-clearing activity of corals and eventual loss/breakdown of cilia cells can also be a consequence of continued exposure to high levels of sediment stress (*Stafford-Smith & Ormond, 1992*; *Stafford-Smith, 1993*; *Erftemeijer et al., 2012*). Given the acute nature of the stress in the Hawai'i experiment (*i.e.*, sediment was added at day 0 and day 4), the corals here may have exhausted their ciliary action abilities, thus leading to downregulation of microtubule-based processes.

## Responses to sterilized white sediment in Florida

No specific responses were shared among *A. cervicornis*, *M. cavernosa*, and *O. faveolata*, despite being exposed to the same sediment for the same amount of time. Given the length of time of this experiment, it is possible that we only captured the longer-term stress responses of these corals and that their responses may have been more similar at the beginning of the exposures. Morphology also may have contributed to the divergent responses between these species. In previous sediment stress studies, morphology contributed to physiological response variability (*Stafford-Smith, 1993*; *Fabricius, 2005*; *Erftemeijer et al., 2012*). For example, *Rogers (1979)* evaluated the effect of shading (as a proxy for turbidity) for 5 weeks on several coral species from San Cristobal Reef, Puerto Rico. The branching coral, *A. cervicornis*, had entirely bleached, while the massive coral, *M. cavernosa*, was visibly unaffected and appeared to have little response. On the other hand, *M. annularis* had substantial bleaching after 5 weeks, despite being a close relative of

*M. cavernosa* (*Rogers, 1990*). Different branching coral species exhibit a wide range of sediment tolerances; after 12 weeks of exposure, the lowest sediment treatments that caused full colony mortality were 30 mg/L$^{-1}$ for *M. aequituberculata* and 100 mg/L$^{-1}$ for *A. millepora* (*Flores et al., 2012*). *Stafford-Smith (1993)* examined sediment rejection efficiency in 22 species of Australian corals from a range of morphologies, finding that there was a wide range of active-rejection efficiencies between species. For instance, *Gardineroseries planulata* is a competent rejector of a variety of sediment sizes, but only for a short period of time. *Favia stelligera* and *Leptoria phrygia* had moderate clearance rates, but tissue mortality occurred within one to two days. *M. aequituberculata* and *Porites* spp. had low rejection efficiency and had bleached tissues, but no tissue mortality for up to 8 days. Morphology was a driving factor in those results, as branching corals had faster clearing rates than massive corals (*Stafford-Smith, 1993*). Thus, it is likely that morphology played a role in driving differences in long-term response in the Floridian corals, highlighting the challenge of predicting responses to sedimentation across species.

Despite differing morphologies, the branching coral, *A. cervicornis*, shared responses related to developmental processes and signal transduction with both massive corals, *M. cavernosa* and *O. faveolata*, independently. Interestingly, no specific responses were shared between *M. cavernosa* and *O. faveolata*, despite similar morphologies. Downregulation of Rho protein signal transduction was observed in *A. cervicornis* and *M. cavernosa* (with the exception of upregulation in the *M. cavernosa* T2vT3 treatment comparison). Rho proteins are part of a superfamily of signaling GTPase proteins, which typically control the assembly and organization of the cytoskeleton, as well as participate in functions such as cell adhesion, contraction, migration, morphogenesis, and phagocytosis (*Mackay & Hall, 1998*; *Moon & Zheng, 2003*; *Phuyal & Farhan, 2019*). In corals, Rho GTPases participate in cytoskeleton remodeling during phagocytosis, as well as cell division of endosymbionts within symbiotic gastrodermal cells (*Li et al., 2014*). Rho GTPase pathways have been identified in coral polyp bailout responses to heat stress and hyperosmosis, as well as in bleached corals in response to low flow environments (*Chuang & Mitarai, 2020*; *Fifer et al., 2021*; *Gösser et al., 2021*). In this study, downregulation of Rho protein pathways suggests that minimal cytoskeleton maintenance, assembly and organization is occurring and that the corals may not be able to properly maintain their cellular structures under sedimentation.

*A. cervicornis* and *O. faveolata* both downregulated chondrocyte development, a developmental response. Chondrocytes are cells in cartilage that make up the cytoskeletal matrix in humans and other animals, including some marine invertebrates (*Philpott & Person, 1970*; *Cowden & Fitzharris, 1975*; *Libbin et al., 1976*; *Archer & Francis-West, 2003*; *Kamisan et al., 2013*). In corals, the cytoskeleton matrix is made up of calcium carbonate, as opposed to cartilage, which forms through rapid accretion of protein rich skeletal organic matrix and extracellular calcium carbonate crystals to form a stony skeleton (*Vandermeulen & Watabe, 1973*; *Akiva et al., 2018*). Skeletal matrix formation begins when planktonic coral larvae settle and begin to secrete calcium carbonate, which helps to anchor the coral to the substrate (*Akiva et al., 2018*). The skeleton grows as the coral animal continues to secrete calcium carbonate, building up a large and intricate 3D skeletal

structure (*Tambutté et al., 2011*). Sedimentation can hinder coral skeletal growth by depositing sediment on the tissue and diverting energy away from growth, as well as decreasing the amount of light that reaches the coral, thus affecting the possibility for light-enhanced calcification, which is responsible for most of the skeletal growth in corals (*Dodge, Aller & Thomson, 1974*; *Erftemeijer et al., 2012*). Downregulation of chondrocyte development may be related to the disruption of skeletal matrix formation, which may influence skeletal density and growth. Decreased skeletal density was observed in corals from inshore reefs which experience higher levels of sedimentation as compared to corals from offshore reefs (*Lough & Barnes, 1992*). Therefore, the downregulation of genes relating to chondrocyte development across morphologies, suggests that corals with a range of morphological characteristics may have decreased skeletal growth in response to sediment stress.

## Differences in response based on morphological characteristics

This study combined data from two independent experiments in order to characterize the molecular mechanisms that corals use to respond to different sedimentation stressors. The experiments used different sediment types (unsterilized red clay sediment in Hawaiʻi, sterilized carbonate sand sediment in Florida) and lengths of exposure (up to 7 days in Hawaiʻi, 18 days in Florida). Differences in experimental methodology prevent us from making direct statistical comparisons between the two experiments; however, the shared morphologies between experiments enables us to identify broad generalizations about conserved gene regulation in response to sediment stress. These comparisons are relevant, as knowledge of what genes and biological processes are broadly affected by sediment stress can help coral reef management.

We did not identify a generalized response across morphology nor gene expression patterns across taxa. There were two groups of three species (*M. capitata*, *M. cavernosa*, *P. acuta* and *A. cervicornis*, *M. capitata*, *M. cavernosa*) that shared specific responses. However, no group contained a single morphology and gene expression patterns varied among species. For example, *M. capitata* (intermediate), *M. cavernosa* (massive), and *P. acuta* (branching) all expressed genes relating to riboflavin transport, the transport of certain vitamins in cells, though they had very different expression patterns. Riboflavin transport genes were upregulated in *P. acuta*, but downregulated in *M. cavernosa*; *M. capitata*, on the other hand, differentially expressed two genes relating to riboflavin transport, one of which was upregulated and the other downregulated. Thus, even though a shared response was identified, the direction of the response varied greatly. This result demonstrates that responses may be shared across morphologies, locations and sediment types, but it may be difficult to predict the directionality of the response. This response may also be due a level of acclimation attained by *M. cavernosa* during the longer exposure in the Florida experiment. The other group, which contained *A. cervicornis* (branching), *M. capitata* (intermediate), and *M. cavernosa* (massive), all downregulated genes relating to positive regulation of skeletal muscle tissue development. This term refers to the activation, maintenance, or increase of the rate of skeletal muscle tissue development (*Buckingham et al., 2003*; *Grefte et al., 2007*). Given that this term is downregulated, it

means that there is little to no activation, maintenance, or increase of the rate of muscle tissue development in these corals. These gene expression patterns highlight the complexity of characterizing responses to different kinds of sedimentation stress in species with different morphotypes.

Some terms were shared across both morphology and location, indicating a generalized sediment stress response. For example, DNA methylation-dependent heterochromatin assembly was shared between *A. cervicornis* (branching) and *P. lobata* (massive). Opposite expression patterns were again observed, in which upregulation occurred in *A. cervicornis* and downregulation in *P. lobata*. DNA methylation-dependent heterochromatin assembly refers to repression of transcription by DNA methylation leading to the formation of heterochromatin (*Jones & Wolffe, 1999*; *Grewal & Moazed, 2003*). In the case of *A. cervicornis*, upregulation suggests that repression of transcription by DNA methylation and subsequent heterochromatin formation is occurring. Therefore, certain portions of DNA cannot be accessed, giving *A. cervicornis* more stringent control on gene expression. On the other hand, the downregulation of these genes in *P. lobata* means less repression of transcription by DNA methylation occurring and heterochromatin is not being fully formed, leaving much of the DNA accessible to transcription machinery and ultimately, reducing the amount of control that *P. lobata* has on gene expression. To date, no work has examined how sediment stress affects epigenetic mechanisms, such as DNA methylation, in coral. However, previous studies have found changes to DNA methylation in response to stress and environmental change (*Putnam, Davidson & Gates, 2016*; *Liew et al., 2018*; *Dimond & Roberts, 2020*; *Rodríguez-Casariego, Mercado-Molina & Garcia-Souto, 2020*). Additionally, it has been observed in corals that genes with weak methylation signals are more likely to demonstrate differential gene expression (*Dixon, Bay & Matz, 2014*; *Entrambasaguas et al., 2021*). Epigenetic modifications and their regulation of gene transcription are highly species and context dependent. Furthermore, the directionality of epigenetic regulation on gene expression or repression can vary depending on the underlying genetic machinery and the environment. This is exemplified in the two species with overlapping response terms. Namely, *A. cervicornis* exhibits a more regulated control on gene expression in contrast to *P. lobata*, which exhibits a less regulated profile of gene regulation. Thus, it is likely that sedimentation stress from each location impacted DNA methylation and heterochromatin in different ways, causing opposing expression patterns.

More general responses were shared over morphology and location, as identified by the orthogroup analysis. The orthogroup analysis grouped homologous gene sequences in different species related to one another by linear descent. The resulting 'orthogroup' represents a group of similar gene sequences across multiple species (*Emms & Kelly, 2015*, *2019*). Orthogroups were shared across morphology and location, though in relatively low numbers (sharing between one and three orthogroups). This sharing may represent a group of orthogroups that form a core group of genes in response to sediment stress. Although we cannot directly compare the genes or orthogroups between experiments, the shared orthogroups represent potential overlap in sedimentation response over location and morphology.

The branching corals, *A. cervicornis* from the Florida experiment and *P. acuta* from the Hawai'i experiment, shared a metabolic response, 'Mo-molybdopterin cofactor biosynthetic process', which describes the creation of the Mo-molybdopterin cofactor, an essential component for catalytic activity of certain enzymes (*Kisker, Schindelin & Rees, 1997*; *Mendel, 2013*). Molybdenum (Mo) is a trace metal synthesized *de novo* through GTP-based processes; cyclic pyranopterun monophosphate (cPMP) is initially formed, which is then converted to the molybdopterin cofactor (*Mendel, 2013*). This essential cofactor catalyzes the oxidation and reduction of molecules in enzymatic processes regulating nitrogen, sulfur, and carbon (*Daniels et al., 2008*; *Iobbi-Nivol & Leimkühler, 2013*). Similar to other results in the present study, both species demonstrated opposing differential gene expression for this term. Mo-molybdopterin cofactor biosynthetic process was downregulated in *A. cervicornis*, but upregulated in *P. acuta*, suggesting that while different sediment types and exposure durations can induce similar differentially expressed genes, it can produce different expression patterns for those genes. Molybdopterin are co-factors for oxidoreductases, a family of enzymes that catalyze the transfer of electrons between molecules (*Kisker, Schindelin & Rees, 1997*). Upregulation of molybdopterin synthesis may suggest that these types of enzymes are more metabolically active. Stressful conditions have made these kinds of enzymes more active in plants and corals (*Bouchard & Yamasaki, 2008*; *Zdunek-Zastocka & Sobczak, 2013*). Upregulation of metabolism related genes was also observed in a study that examined transcriptomic responses of corals in response to two different sediment experiments (*Bollati et al., 2021*). Downregulation of Mo-molybdopterin cofactor synthesis may indicate that the coral does not have enough energy to synthesize molybdopterin which in turn makes it so the activity of these specific enzymes is decreased or stopped altogether, suggesting a decrease in metabolism for *A. cervicornis*. It is also possible that the unsterilized sediment was providing molybdopterin to *P. acuta*, making it necessary for *P. acuta* to upregulate genes relating to molybdopterin processing to manage the influx (*Fujimoto & Sherman, 1951*; *Siebert et al., 2015*). Because the sediment was sterilized in the Florida experiment, no molybdenum would be present in the sediment, indicating that molybdenum-related enzymes may not have been functioning at a high level, leading to downregulation.

## CONCLUSION

This study incorporated data from two separate experiments to more fully characterize the molecular mechanisms induced by sedimentation in corals. We found that developmental processes are primarily impacted in branching corals across location, highlighting potential future research avenues with regards to sediment stress and reproductive potential and output. While few specific genes were shared across morphology and location, orthogroup analysis uncovered potential overlap in generalized sediment stress responses. Direct comparisons across species are necessary to further elucidate the genetic basis of coral susceptibility to sediment stress.

# ACKNOWLEDGEMENTS

We thank Erin Chille for assistance with bioinformatic analyses. We thank the Hawaiʻi Institute of Marine Biology (HIMB) in Kāneʻohe Bay, Oʻahu, Hawaiʻi and the Smithsonian Marine Station (SMS) in Fort Pierce, Florida for facilities support.

### Funding

This study was supported by the National Fish and Wildlife Foundation, NSF HDR Awards #1939795 and #1939263, and an NSF GRFP. The Tufts Data Intensive Studies Center (Tufts DISC) supported part of the APC for this article. The funders had no role in study design, data collection and analysis, decision to publish, or preparation of the manuscript.

### Grant Disclosures

The following grant information was disclosed by the authors:
National Fish and Wildlife Foundation, NSF HDR: #1939795 and #1939263, and an NSF GRFP.
The Tufts Data Intensive Studies Center (Tufts DISC).

### Competing Interests

The authors declare that they have no competing interests.

### Author Contributions

- Jill Ashey performed the experiments, analyzed the data, prepared figures and/or tables, authored or reviewed drafts of the article, funding acquisition, and approved the final draft.
- Hailey McKelvie analyzed the data, authored or reviewed drafts of the article, data validation, and approved the final draft.
- John Freeman analyzed the data, authored or reviewed drafts of the article, data validation, and approved the final draft.
- Polina Shpilker analyzed the data, authored or reviewed drafts of the article, data validation, and approved the final draft.
- Lauren H. Zane analyzed the data, authored or reviewed drafts of the article, and approved the final draft.
- Danielle M. Becker analyzed the data, authored or reviewed drafts of the article, and approved the final draft.
- Lenore Cowen analyzed the data, authored or reviewed drafts of the article, resources, funding acquisition, and approved the final draft.
- Robert H. Richmond conceived and designed the experiments, performed the experiments, authored or reviewed drafts of the article, resources, funding acquisition, and approved the final draft.

- Valerie J. Paul conceived and designed the experiments, performed the experiments, authored or reviewed drafts of the article, resources, funding acquisition, and approved the final draft.
- Francois O. Seneca conceived and designed the experiments, performed the experiments, authored or reviewed drafts of the article, resources, funding acquisition, and approved the final draft.
- Hollie M. Putnam analyzed the data, authored or reviewed drafts of the article, resources, funding acquisition, and approved the final draft.

### Field Study Permissions

The following information was supplied relating to field study approvals (*i.e.*, approving body and any reference numbers):

Hawaiian corals were collected in Kāneʻohe Bay, Oʻahu, Hawaiʻi under Hawaiʻi SAP permit (SAP 2015-48). Floridian corals were collected from the Key West nursery of the Florida Keys National Marine Sanctuary under NOAA National Marine Sanctuaries Permit #FKNMS-2015-016 and #FKNMS-2016-017.

### DNA Deposition

The following information was supplied regarding the deposition of DNA sequences:

The raw sequences are available at NCBI BioProject: PRJNA911752.

### Data Availability

The code for analyses is available at GitHub and Zenodo:

- https://github.com/JillAshey/SedimentStress.

- Ashey, J. (2023). SedimentStress [Data set]. In PeerJ. Zenodo. https://doi.org/10.5281/zenodo.10183389.

- https://github.com/JillAshey/FunctionalAnnotation.

- Ashey, J. (2023). Functional annotation [Data set]. In PeerJ. Zenodo. https://doi.org/10.5281/zenodo.10211582.

The archived references used in this study are available at OSF: Ashey, Jill, and Hollie Putnam. 2022. "Sediment Stress." OSF. December 27. https://doi.org/10.17605/OSF.IO/8QN6C.

The raw sequences are available at NCBI BioProject: PRJNA911752.

### Supplemental Information

Supplemental information for this article can be found online at http://dx.doi.org/10.7717/peerj.16654#supplemental-information.

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
