# Peer review of "Characterizing transcriptomic responses to sediment stress across location and morphology in reef-building corals"

_PeerJ, doi:10.7717/peerj.16654_

## Round 0.1 · original submission · Minor Revisions

The reviewers have each provided comprehensive suggestions for editing of the text and highlight some regions where the authors can assist readers in understanding the study.

The authors are supportive of the study and praise many aspects of the paper, including reproducibility of the study and open sourcing of the associated data.

Reviewer 1 ·

Basic reporting

Please see additional comments for all review comments.

Experimental design

Please see additional comments for all review comments.

Validity of the findings

Please see additional comments for all review comments.

Additional comments

The manuscript by Ashey et al. describes two interesting and novel sedimentation studies using tropical reef-building corals. The authors aim to assess different molecular responses of these corals under sedimentation stress and make these comparisons across different morphologies. While the study provides some really great insights into this important area of work, the manuscript could use some additional work to improve the readability and clarity that I have outlined below. Overall, the authors do a fantastic job providing a review of the relevant literature and comparing their study to previous work and I really applaud the authors for their support of open access data/code/science.


Introduction
While your introduction is a very thorough review of the literature, it is a bit length so I recommend you considering condensing it to cover only the most critical background to set up your study. Some of the more specific details could be moved into your discussion as you cover those specific processes.

Methods
Please introduce early on (in that intro paragraph) and very clearly how you will differentiate between your two different studies. If you will refer to them as Hawai`i and Florida, make that clear for your readers.

At some point (maybe not necessarily in the methods) please make sure you describe your reasoning behind the different sediments types. This is important design considerations that will help readers follow your study better.

Regarding your comment about comparability, I do think you need to treat these as completely separate experiments statistically since they are different species, performed at different times, and use different durations.

For both experiments, please provide details of the water quality throughout the experiment (i.e., salinity, pH, temperature, etc) and provide the light levels and total light time per day. Also please describe any water changes, if the systems were flow through (or where the water was sourced from), feeding details, etc. These details are critical for understanding the conditions the corals were exposed to throughout the experiment that may impact the molecular responses to your treatment conditions.

Love the open access data/code repository! I do recommend trying to make your README informative about where to find the different data/scripts necessary to repeat your analyses to promote cade sharing.

Please make sure you appropriately cite R (include the citation, not just the version).

Results
Overall, I found several sections of the results a bit difficult to follow. Sections where you present several different numbers and then define them later are tough to follow (for example: “A. cervicornis, M. cavernosa, and O. faveolata, had 278, 158, and 27 unique GO terms assigned to their DEGs, respectively”). I suggest you rewrite these to clearly link the values with the species. Additionally, some shorter sentences may help your readers to follow your results here since you have a lot of data to present. It may also help to split your results into the two sections of your two different experiments since they do have very different methods.

In places where you discuss ‘dropping’ or ‘pooling’ data because there were no differences, please provide appropriate statistical backing for these decisions.

Again, I am unsure about the comparison across different experiments/species as an appropriate analysis. I think you have a lot of value in your experiments as two separate studies that can be qualitatively compared, but not statistically. If you feel this concern is in error, you will need to provide significant references supporting your decision here as I will not be the only reader with this concern. Please check to see if it is actual gene-sharing, rather than just GO term sharing, that you are seeing here. I think this would be a bit stronger.

I am more comfortable with your orthogroup comparison, but again feel you need to be extremely careful with this comparison and the conclusions you make on two very different experiments.

Discussion
Your statement in this first paragraph “The methodological differences prevent us from making direct statistical comparisons between the experiments.” Seems to counter what you present in your introduction and the methods. If you are not performing direct statistical comparisons, please make that clear early on to prevent concerns (like mine as stated above) about your methods. This first paragraph also feels like a better fit to put at the end of your intro to set up your study than where it is in the discussion.

In general, your discussion would benefit from a bit more concise writing. Some of your discussion include methods details that are unnecessary and presented previously. While you want to make sure you appropriately acknowledge the work that was done previously (which you have done a fantastic job of), you also want to make sure your discussion doesn’t hide your own study. I think you discussion could use some thoughtful reworking of the text to shorten it and make sure your conclusions are not being hidden too much by your review of previous studies.

Figures
1: nice job creating a clear experimental overview. This is really important when presenting multiple experimental designs

2: I recommend making the font in this figure a bit larger to improve readability

3: Again, make font larger. There is also a lot of white space on this figure and I wonder if there is a way to move things around to make better use of this space. Same comments for figure 6

Reviewer 2 ·

Basic reporting

This is in general a well-written paper, but authors have combined two independent experiments with different designs which has made interpretations in the discussion section more complex and harder to follow. The discussion could be condensed as it is a very long discussion. I would still consider publishing the two studies separately and think that further justification for combining the two studies is needed. I understand that publishing studies separately may provide repetition in some findings but the interpretations of the results would be more simple and robust. In this case, analyses could do a deeper drive into functionally annotated DEG given that GO enrichment terms can usually be a bit ambiguous especially for non-model organisms. Results from the 18 day experiment appear to show a greater extent of acclimation of the treatments given that all sediment treatment gene profiles are similar while the 7 day experiment shows clustering by treatment (PCA plots); this should be discussed. From my knowledge, all appropriate literature references are covered. Raw data and code is provided. Authors are aware of the limitations of their study (comparing two independent studies with differing designs) and provide caution with their interpretations/conclusions.

Experimental design

The two independent studies reported in this manuscript are original experiments and provide new data on long term sedimentation stress on multiple coral species from two locations. This analysis can only provide a generalised comparison between morphologies and locations given the lack of consistency between the two studies' design however, the two studies are both interesting and valid for publication in their own right.

From a technical point this study performs a standard RNA-Seq analysis and reports methods with a good understanding. Authors provide detailed scripts on Github for reproducing analyses. Details of both studies included are relatively well-documented however, justifications for a few experimental design aspects (e.g. sterilised versus unsterilised sediment types and water turn over, see specific comments below) need clarification.

Validity of the findings

Authors clearly state their awareness to the limitations of comparing 2 independent experimental designs in the methods and discussion and highlight how the comparison is a more generalised one comparing coral morphologies and location to relatively long term sediment stress. Both individual studies are well designed and novel in their own right. For the RNA-Seq analysis, the lowest replication of n=3 is low but sufficient for such an analysis. All raw data and coding is accessible. Conclusions are limited to supporting results. This paper suggests that sedimentation stress may negatively affect development of corals and their reproductive potential. Authors highlight the drawbacks of their comparisons (comparing different species in the two studies within groups of morphologies) by indicating how direct comparisons across species are necessary in future studies as there was a relatively weak general response under long term (7 or 18 days) sedimentation between groups of different morphologies and stronger species-specific responses. Overall, I would reconsider the publication of this manuscript once combination of the two studies is further justified and specific comments below (see additional comments section) are addressed.

Additional comments

Specific comments:
Line 34 correct ‘Differentially’ to ‘Differential’ and remove (DEGs) as that typically denotes ‘differentially expressed genes’.
Line 74 consider splitting this long sentence into two for readability.
Line 145 consider adding additional reference supporting upregulation of HSPs in coral under deoxygenation stress (Alderdice et al. 2021)
Line 211 correct ‘concentrations’ to ‘concentration’
Line 215 provide references supporting how relevant the low and high sediment concentrations are. Also, you mention powerheads were attached to the bottom of tanks to ensure water motion but was there water changed at all for either study, please clarify. If not, please justify why as longer term experiments will be subject to waste accumulation.
Line 231 correct number order from ’15-12L’ to ’12-15L’, were tanks not all the same volume? Please justify why as water volume to coral biomass ratio can have an impact on stress responses.
Line 231 clarify how you get n=5 from using 3 replicate tanks and only 1 fragment per tank. Would it not be n=3?
Line 236 justify why the sediment was sterilised? One could assume that the associated microbiome of the sediment is very important in dictating an organism’s stress response as also mentioned in discussion Line 496. Also, please clarify why the silt-clay proportion of the sediment was filtered out, seeing as silt seems to be problematic for corals during sedimentation as you mentioned in discussion Line 493.
Line 269 specify how RNA was quality checked and what was the passing threshold.
Line 273 great to see scripts accessible on GitHub, thank you!
Line 325-7 great to see a power analysis done but I am unable to find the filtered gene set in Table S1 or the standard error of the mean range which ranged from 0.012 to 0.06. Please provide or clarify location.
Line 381 number of DEGs are generally quite low but this may reflect a level of acclimation to the sediment stress given how long the stress lasts for? Consider adding this comment.
Line 388 when you refer to ‘comparing all treatments’, do you combine all treatments into one group and compare against control group? Or are they separate comparisons between each treatment and control?
Line 407 are there more GO enriched terms that DEGs, assuming the same DEG appears in multiple enriched GO terms? Also when looking at the DEGs do you find any hypoxia stress associated genes? as other sedimentation studies such as Bollati et al. 2021 has indicated hypoxia to drive the transcriptional response in corals.
Line 480 paragraph starting at this line, suggests that you will have a integrated discussion on these studies but you discuss them separately. Please stated that to help readers anticipated the discussion layout given that the results had an integrated format of the two studies.
Line 511 consider removing ‘In our study’ to improve the flow of the sentence with the previous one.
Line 517 it is true that the functional gene analysis on vertebrate model organisms therefore in such cases is it helpful to report some specific DEGs of the enriched GO terms to get an idea of what is actually going on as functionally annotated processes can be less relevant to coral (i.e. non-model organisms) as you already mention. Consider describing some important genes from these processes which have been previously reported in coral or are key conserved metazoan genes.
Line 552 this response could represent a longer-term response to sedimentation stress as it may be upregulated in the short-term response. Consider mentioning the timing of the stress response.
Line 663 rephrase to make it more relevant to corals as corals do not possess skeletal muscle tissue.
Line 715 the different directions of certain gene regulation could be due to the different timing of stress response, consider mentioning this.
Figure 1 consider changing font colour to white for those with a dark colour background to improve readability.

References:
Alderdice, Rachel, David J. Suggett, Anny Cárdenas, David J. Hughes, Michael Kühl, Mathieu Pernice, and Christian R. Voolstra. 2021. “Divergent Expression of Hypoxia Response Systems under Deoxygenation in Reef‐forming Corals Aligns with Bleaching Susceptibility.” Global Change Biology 27 (November): gcb.15436.
Bollati, Elena, Yaeli Rosenberg, Noa Simon-Blecher, Raz Tamir, Oren Levy, and Danwei Huang. 2021. “Untangling the Molecular Basis of Coral Response to Sedimentation.” Molecular Ecology, November. https://doi.org/10.1111/mec.16263.

Reviewer 3 ·

Basic reporting

This study presents a comprehensive analysis of two experiments examining the effect of sedimentation on six coral species with different growth forms. The authors openly address the discrepancies between the two experiments and interpret the results in the context of each experiment and overall. The studies appear to be soundly conducted, and the relevant methods are stated in the text. In general, the manuscript seems a bit too detailed in places, which distracts from the main message and question. Both the introduction and the discussion could be more focused.

Experimental design

One aspect that could greatly improve the excellent dataset presented here is considering an analysis based on quantifying growth forms instead of categorizing them. For example, a study by Zawada et al. (https://link.springer.com/article/10.1007/s00338-019-01842-4) demonstrates the benefits of quantifying growth forms. This approach could provide a better understanding of the impacts of growth forms and help identify which traits of the growth form correlate with the corals' responses to sedimentation. Additionally, using a more recent method to address coral growth forms would align with state-of-the-art methods used to analyze the corals' responses. If coral skeletons of the tested organisms are not available for quantification, openly available 3D models of the studied species could be used to derive metrics.

Validity of the findings

No comment

Additional comments

L172: Pocillopora acuta (formally Pocillopora damicornis) formally or formerly
L489-499: this is a justification of the study, can be moved to the methods and also some of it is redundant with the methods
L555- 566: see comment above
Tables: Improve table designs to increase readability (check linebreaks and adhere to scientific convention for tables e.g. filling, lines, text left aligned, numbers right aligned)
Figures 4&5: Are the total counts of GO terms biased given the unequal sample number belonging to the 3 different categories for morphologies (branching: 2, intermediate:1 and massive:3)/locations?
Discussion: The discussion analyzes the results in great depth. The extensive species-specific discussion from line 480 to 632 (more than six pages) could be reduced to better focus the message of the paper. The paragraph titled 'Differences in response based on morphological characteristics' may be more aligned with the hypothesis and title of the paper and could serve as the main focal point.

---

## Round 0.2 · accepted · Accept

Thank you for making the changes to the manuscript. There are a couple of minor suggestions made by one of the reviewers that can be taken care of during production.

Reviewer 2 ·

Basic reporting

I still think that these two experiments would benefit from being published separately and assessed at a more fine scale level given that they present RNASeq data which is very sensitive to differences in timing of sampling and the inconsistencies in responses would be greatly influenced by the multiple differences between studies. However, Authors have now updated the manuscript to make sure that the two experiments are separately described, only qualitatively compared and limitations are well-highlighted. On this basis I would accept this publication after addressing the few minor revisions.

Experimental design

No further comment

Validity of the findings

Regarding the use of sterilised sediment in the Florida experiment (lines 284-299 and 303-307), I think the extent of ecological relevance still needs to be specified as it could reflect the mechanical stressor but lacks the associated biological stressors such as the accompanying microbiome.

Additional comments

Refer to TAE abbreviation in full (Lines 344-347).

Correct 'diuerentially' to 'differentially' in Figure 2 legend

Reviewer 3 ·

Basic reporting

I am glad to see that the authors have addressed all my previous comments, and I congratulate them on their great manuscript.

Experimental design

No additional comments.

Validity of the findings

No additional comments.

Additional comments

No additional comments.